# Effects of Modified Cellulose Fiber and Nanofibril Integration on Basic and Thermo-Mechanical Properties of Paper

Ayyüce Güzide Teke [1,*,†], Celil Atik [2], Jani Bertoncelj [3,4], Ida Poljanšek [4] and Primož Oven [4,*]

1 Department of Forest Products Chemistry and Technology, Institute of Graduate Studies, Istanbul University—Cerrahpasa, Valide Sultan St. Nu:2, Sarıyer, Istanbul 34473, Turkey
2 Department of Forest Products Chemistry and Technology, Faculty of Forestry, Istanbul University—Cerrahpasa, Valide Sultan St. Nu:2, Sarıyer, Istanbul 34473, Turkey; celilatik@iuc.edu.tr
3 Labtim d.o.o., Ziherlova Ulica 6, 1000 Ljubljana, Slovenia; bertonceljani@gmail.com
4 Department of Wood Science and Technology, Biotechnical Faculty, University of Ljubljana, Rožna Dolina, Cesta VIII/34, 1000 Ljubljana, Slovenia; ida.poljansek@bf.uni-lj.si
* Correspondence: ayyuceguzidegok@gmail.com (A.G.T.); primoz.oven@bf.uni-lj.si (P.O.)
† This manuscript is a part of the doctoral thesis entitled "The Effect of Short and Long Fibers on Thermo-Mechanical Properties of All Cellulose Composite".

**Abstract:** This study investigates the influence of fiber modification methods and beating degrees on the properties of paper sheets. Two different methods were used to modify fibers: NaOH + urea and TEMPO ((2,2,6,6-tetramethylpiperidin-1-yl)oxidanyl) and blended with traditional paper fibers. Subsequently, we evaluated the resulting sheets for their optical, strength, and thermo-mechanical characteristics. Notably, we also scrutinized sheets created exclusively with 100% TEMPO-modified fibers. The addition of modified fibers led to improvements in several strength properties, but it had a noteworthy negative impact on the optical properties of TEMPO-treated fibers compared to the other papers. Furthermore, thermal analysis revealed that the contraction rates of the samples increased considerably up to 40–50 °C for the out-of-plane direction and surpassed 130 °C for the in-plane direction. In general, the inclusion of modified fibers had a significant effect on thermo-mechanical properties. Specifically, TEMPO modification resulted in an increase in the maximum in-plane contraction ratio, shifting it from −0.40% to −0.59%, along with its corresponding temperature. This research underscores the potential of modified fibers to enhance paper properties and contribute to the development of more sustainable paper-based products.

**Keywords:** modified fibers; nanocellulose; TEMPO; mechanical properties; optical properties; thermal contraction

## 1. Introduction

Paper production is the most widespread utilization area of cellulosic fibers. The demand for cellulosic materials with improved properties has prompted researchers to investigate the modification of cellulosic fibers [1,2]. Modified cellulosic fibers are cellulose fibers that have undergone various chemical or mechanical treatments to enhance their properties. Cellulose fibers can be modified in several ways. One of the common modification systems is mechanical treatment like refining and beating.

The mechanical fibrillation process consumes a significant amount of energy. Therefore, to lower the energy consumption of mechanical disintegration of pulp fiber, chemical or enzymatic pretreatments have been recommended [3]. Chemical treatments like bleaching, sizing, and surface coatings, as well as mechanical treatments, are performed to alter the fiber's physical, chemical, and surface properties. Modified cellulosic fibers can improve the paper's strength and durability. This is especially important in applications where the paper needs to withstand mechanical stress, such as packaging materials or certain printing applications. Surface modifications can enhance the printability of paper by reducing ink

absorption, improving ink holdout, and ensuring sharper image reproduction. This is crucial in the production of high-quality printing papers and magazines. The addition of modified fibers can help control the porosity of paper, affecting its permeability to liquids and gases. This can be important in applications like filter papers, where precise filtration properties are required. Surface treatments can influence the paper's optical properties, such as brightness, opacity, and color. These factors are crucial for applications like photographic papers, art papers, and specialty printing. The choice of modified cellulosic fibers can also impact the environmental footprint of paper production. Using recycled fibers or sustainably sourced fibers with minimal chemical treatments can lead to more ecofriendly paper products. Modified cellulosic fibers are often used in the production of specialty papers for specific applications, such as security papers (e.g., banknotes), thermal papers (e.g., receipts), and specialty packaging materials. Ongoing research in the paper industry aims to develop new and innovative ways to modify cellulose fibers to meet evolving market demands, including sustainability requirements and the reduction in environmental impacts.

The addition of modified cellulosic fibers is a critical aspect of papermaking that allows manufacturers to tailor the properties of paper to meet the specific needs of various applications. These modifications can enhance paper strength, printability, and other performance characteristics while also addressing environmental concerns.

Furthermore, to maintain the high quality of recycled paper products, the recovered materials must be carefully classified [4,5]. Improving the strength and other properties of paper through the addition of modified cellulosic materials may be the most suitable approach from an environmental standpoint. Additionally, cellulosic papers are finding new areas of application, such as electric insulators and electronic devices [6].

One approach to development in this direction is the addition of partially dissolved cellulose to paper fibers [7]. The dissolution of cellulose in an NaOH + urea aqueous solution is one of the most well-known methods [8–10].

It is well recognized that NaOH plays a leading role during the dissolution of cellulose in NaOH + urea aqueous solution. NaOH can penetrate not only between crystallites but also into the crystallites, thereby destroying the inter- and intrahydrogen bonds between cellulose molecules. On the other hand, urea functions as a hydrogen bond donor and receptor, preventing the reassociation of cellulose molecular chains [11].

The possible swelling mechanism for the NaOH + urea application is as follows: NaOH hydrates destroy the intermolecular and intramolecular hydrogen bonds in cellulose, while the amino groups of urea easily form hydrogen bonds with hydroxyl groups in cellulose. The synergistic action of NaOH + urea prevents binding between cellulose molecules and weakens the binding force of cellulose molecules [11].

Another approach to improve the properties of the paper involves increasing the bonding between fibers by adding nanofibrils. The application of cellulose nanofibrils (CNF) or microfibrils can be useful during the papermaking process. Costa et al. [12] showed the positive effects of cellulose nano/microfibrils on the properties such as morphological, crystallinity, physical–mechanical, and air barrier of paper hand sheets despite the fact that no type of cationic polymer is used to assist the retention of micro/nanofibrils and fibers.

The production of nanocellulose is an energy-intensive process, and TEMPO oxidation is a widely used method to decrease energy consumption. TEMPO-modified nanocellulose can be utilized in the paper industry, among many other application areas [13–16].

During TEMPO-mediated oxidation, the nitroxyl radical affects the oxidation from the alcohol to the aldehyde groups, while the hypobromide generated in situ from hypochlorite and bromide performs the further oxidation of the aldehyde groups to the carboxyl groups [17].

Yang et al. [18] described how TEMPO-modified nanocellulose enhances the strength properties of all-cellulose composites.

Papermakers usually use blends of different pulps to obtain paper with the desired properties. Moreover, the fibers are beaten before the papermaking process, which alters their interaction with additives and consequently affects the properties of the paper.

This work focuses on the impact of adding NaOH + urea-modified fibers or TEMPO-modified nanocellulose (TMNC) or to produce TEMPO-modified fibers paper on the strength, optical, and thermo-mechanical properties of paper. A variety of pulp sources has made a demand for having a fundamental process for improvement of the fiber quality that can be applied to all types of pulps. One of the processes that are conducted in the stock preparation is so-called "pulp refining" or "beating". Pulp refining or beating could be described as a mechanical treatment of the pulp by using special equipment. Beating can affect the fiber structure and its properties through some simultaneous changes [19]. Thus, the pulp can obtain the characteristics required by the papermaking process to achieve the desired quality of the paper. Therefore, four different beating degrees (0, 4000, 6000, and 8000 revolutions) were used in the study. The baseline of the manuscript was to determine the paper with optimum strength properties and to perform the thermo-mechanical analyses on it.

## 2. Materials and Methods

### 2.1. Materials

Bleached softwood fibers (spruce, pine, and larch) were obtained from Zellstoff Pöls AG, Pöls, Austria. NaOH, urea, HSO, NaBr, NaClO, and TEMPO chemicals were supplied by Sigma Aldrich (Taufkirchen, Germany).

The fiber furnish was prepared by mixing 70% spruce, 25% pine, and 5% larch pulp. A pulp blend (30 g o.d.) was beaten at a PFI mill (Deerlijk, Belgium) according to the ISO 5264-2 [20] standard method at four different stages: 0, 4000, 6000, and 8000 revolutions.

### 2.2. Modification Methods

The pulp was chemically modified using the following methods: acid hydrolysis and alkaline dissolution (AHAD). Acidic hydrolysis conditions included a fiber consistency of 4%, $H_2SO_4$ (50%), a temperature of 40 °C, and a duration of 10 min [21]. After the acidic hydrolysis process, the pulp was washed until it reached a neutral pH. Partial dissolution in an alkaline medium involved an alkaline solution of 7% NaOH, 12% urea, and 81% water, cooled down to −12 °C. After stirring the solution for a minute, the pulp was slowly added and continuously stirred for 10 min. The pulp was then washed to remove residual chemicals. Ultraturrax (IKA T-25, Staufen, Germany) was used to homogenize the treated material after washing. The resulting material had a semigel form with a dry content of about 2.7%.

TEMPO-modified nanocellulose (TMNC) was produced from 30 g oven-dried cellulose and treated with TEMPO reagent ((2,2,6,6-tetramethylpiperidin-1-yl)oxidanyl) according to the procedure developed by Saito and Isogai [22]. The cellulose was dispersed in 1.8 L of distilled water, and the mixture was stirred at 300 rpm for approximately 20 min. The desired amount of TEMPO and NaBr was added to the suspension, which was stirred for a while. NaClO was then added to the suspension, and the pH was controlled (with NaOH) until it reached a constant value of pH 10. The pulp was washed several times to remove residual chemicals. The cellulose was then homogenized using a GEA Lab Homogenizer (PandaPLUS-2000, Parma, Italy), and after three passes, the material in gel form was obtained.

TEMPO modified fibers (TMF) were obtained by the same procedure as mentioned above, except for homogenization. The pulp was washed several times to remove residual chemicals.

Chemically modified material was added to the beaten pulps at ratios described below:

- No addition of modified fibers.
- AHAD modified fibers 1%.
- AHAD modified fibers 3%.

- TEMPO modified nanocellulose 3%.
- TEMPO modified fibers 100%.

  Paper sheets were prepared according to the Rapid-Köthen method (ISO 5269-2 [23]).

*2.3. Paper Sheet Characterization*

The addition of different substances may cause changes in basic properties of paper. The following ISO standard test methods were used, as well as some references which used the same measurement methods: ISO 5270 (determination of properties of laboratory sheets [24]), ISO 536 (basis weight [25]), ISO 534 (thickness, measured with paper micrometer [26]), ISO 1924-2 (tensile strength, tested with Zwick-Roell 2.5 kN, [27]), ISO 2758 (bursting strength, tested with Zwick-Roell (Ulm, Germany) 2.5 kN, [28]), static tearing resistance (tested at Zwick-Roell 2.5 kN according to Atik and Engin [29]), ISO 2470 (ISO brightness, measured with Elrepho Datacolor (Suzhou, China, [30])), ISO 11476 (CIE whiteness, measured with Elrepho, [31]), ISO 2471 (opacity, measured with Elrepho, [32]), ISO 5631 (yellowness and CIE Lab* color values, measured with Elrepho, [33]), ISO 2493 (stiffness, [34]), and ISO 5636-3 (porosity, [35]).

FT-IR analyses were conducted using a Perkin Elmer 100 FT-IR spectrometer (Beaconsfield, UK) equipped with an ATR unit (Universal ATR Diamond Zn/Se). The FT-IR spectra were collected over the wave number range of 650–4000 cm$^{-1}$.

The thermo-mechanical properties of the samples were determined using a Shimadzu TMA-H60 (Kyoto, Japan). In-plane measurements were performed in elongation mode for 5 mm wide samples with a 20 mm distance between clamps. Z-directional measurements were performed in pressure mode using a stack of 5 paper samples with a 5 mm width. The test conditions included a temperature range from 30 °C to 200 °C, a heating rate of 10 °C·5 min$^{-1}$, air flow, and a constant load of 100 mN. The formula used when calculating the thermal coefficient is as follows:

$$\alpha = \frac{(Ls - L_0)}{(L_0 \times dT)}$$

$\alpha$: coefficient;
$L_S$: final length;
$L_0$: initial length;
$dT$: temperature change.

## 3. Results and Discussions

The basis weight and density properties of the paper sheets are given in Table 1. As expected, the density of the paper increases with the beating degree as well as additional modified fibers.

**Table 1.** The basis weight and density values of papers.

| Modified Fiber Content | Basis Weight (g·m$^{-2}$) | | | | Density (kg·m$^{-3}$) | | | |
|---|---|---|---|---|---|---|---|---|
| **PFI Revolution** | **0** | **4000** | **6000** | **8000** | **0** | **4000** | **6000** | **8000** |
| No modified fibers | 77.4 | 83.9 | 79.8 | 81.4 | 534 | 717 | 726 | 740 |
| 1% AHAD | 81.4 | 87.1 | 82.2 | 81.4 | 509 | 764 | 776 | 791 |
| 3% AHAD | 85.5 | 81.4 | 82.2 | 83.9 | 570 | 727 | 776 | 799 |
| 1% TMNC | 79.8 | 81.4 | 83.9 | 83.9 | 532 | 679 | 784 | 795 |
| 100% TMF | 77.7 | 75.0 | 76.6 | 78.2 | 706 | 750 | 782 | 798 |

Although TMF samples have the lowest basis weight values, TEMPO modification has a significant effect on paper density, with TMF reaching a value of 706 kg·m$^{-3}$ for unbeaten

pulp. The highest density values were obtained for pulp beaten at 8000 PFI rev. All sheets with modified fibers were at least 6.9% higher in density compared to the control sample.

Paper porosity is a property of paper that indicates the penetration of gas or liquids from the surface through it. Some properties, such as runability and printability, are highly dependent on the porosity of paper. The addition of short fibers and fillers, the application of surface sizing, the coating, and the performance of calendaring determine the final porosity of paper products. Commercial office papers have a porosity of about 1320 mL·min$^{-1}$ [22], while the investigated paper samples will reach this level at approximately 3000 PFI revolutions beating (Figure 1).

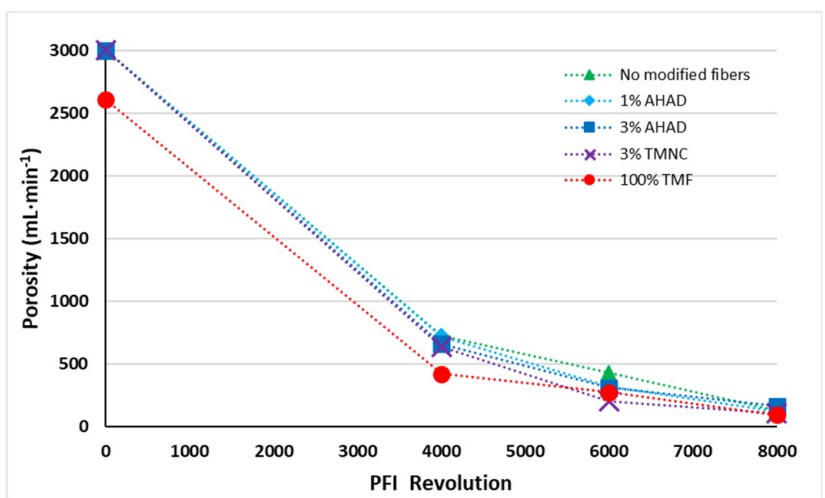

**Figure 1.** Porosity of paper sheets.

The sheets obtained by TEMPO-modified fibers have significant decreases in porosity properties at the nonbeating group and 4000 revolutions (2604, 33, and 420 mL·min$^{-1}$, respectively) compared to the other samples (Figure 1). Furthermore, differentiation between samples decreases with increasing beating. Lower values were observed at 8000 revolutions for all the groups.

Figure 2 illustrates typical spectra of cellulose, with the 850–1150 cm$^{-1}$ region representing the backbone of cellulose. A broad signal peaking around 3300 cm$^{-1}$ is attributed to stretching OH$^{-}$ bonds as well as hydrogen bonds between cellulose chains. The signal at 1336 cm$^{-1}$ corresponds to C-OH bending bonds, while the signal around 1636 cm$^{-1}$ indicates the bending of OH due to absorbed water. The peak that appeared at 896 cm$^{-1}$ is the CH-stretching vibration of the glycoside bond [36]. A reduction in peaks at 896 cm$^{-1}$ and 986 cm$^{-1}$ was observed for paper produced from TMF. Notably, TEMPO-oxidized cellulose is distinguished from other samples by the shift and merging of stretching at 1598 cm$^{-1}$, owing to the presence of the C=O moiety of carboxylate groups on fibrils [37–39]. The carboxylate band in the spectrum of 100% TMF sheets masked the OH bending of adsorbed water at 1635–1638 cm$^{-1}$ [40].

Ma et al. [41] reported a loss of mechanical properties when the carboxyl content increases and suggested that the sheets should contain a limited amount of oxidized cellulose.

The lateral order CI values of the samples show a weak response to the beating process, with values ranging between 0.485 and 0.517. In contrast, the CI of the TMF sheets exhibits higher values, ranging between 0.655 and 0.699, which increase with further beating (as shown in Figure 3). The most significant increase was observed during the initial beating stage, where the removal of amorphous cellulose occurred due to the action of TEMPO treatment of fibers. Zhao et al. [42] indicated that the variability in the cellulose crystallinity during mechanical beating is a hypothesis that has been considered in some studies and remains an uncertainty [19,42]. They also determined in their research that the correlation between the cellulose crystallinity index and the degree of beating was not linear but an

initial upward and then downward trend followed by a repeating fluctuation as a result of the beating action on amorphous regions first and then on crystalline cellulose [42].

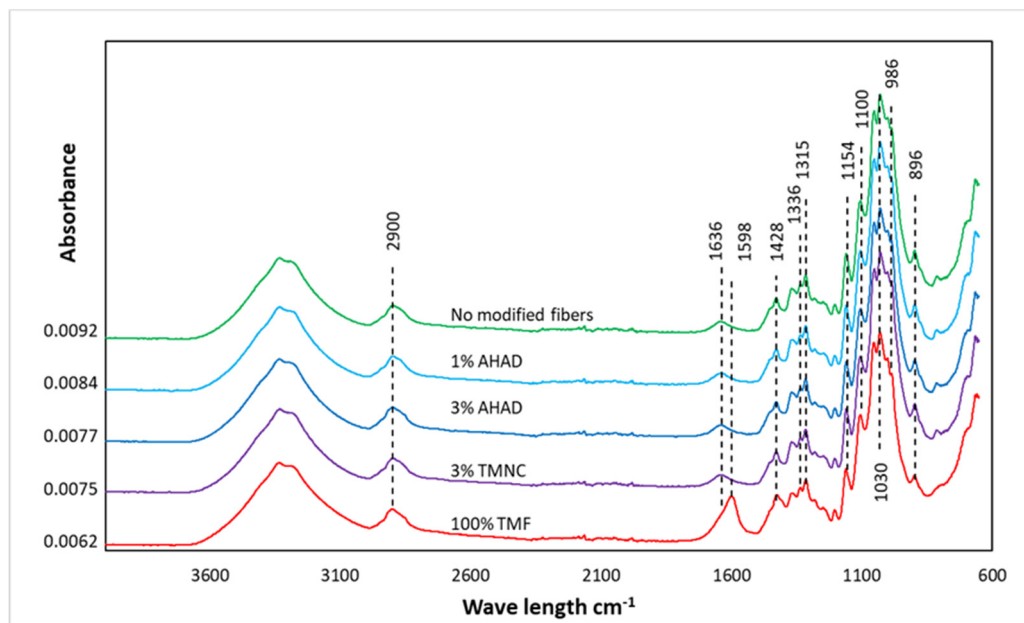

**Figure 2.** FTIR absorbance spectra of paper sheets.

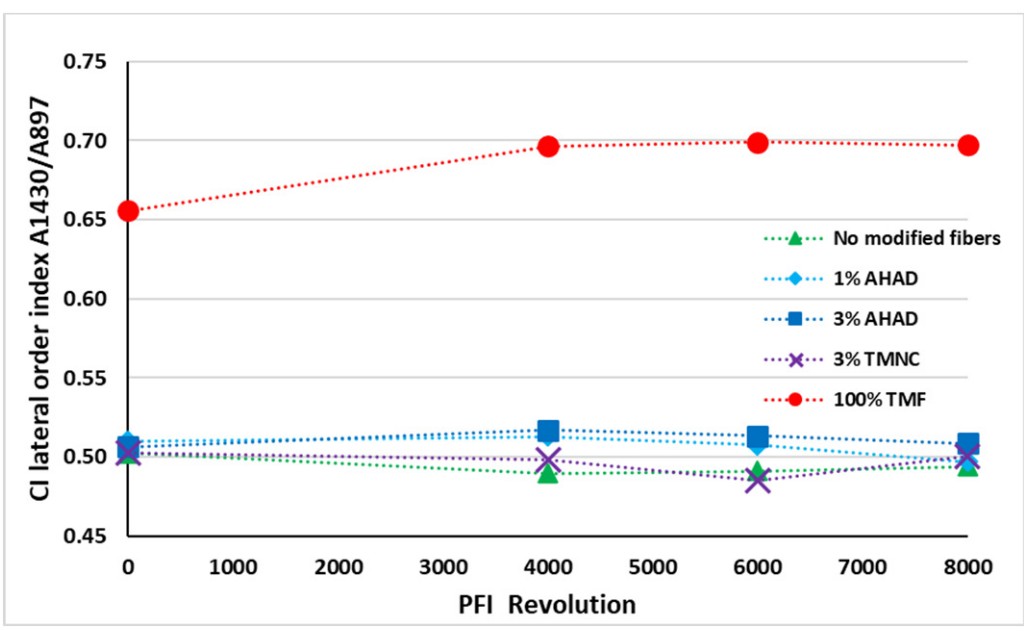

**Figure 3.** Lateral order crystallinity index A1430/A897.

The tensile index of all samples except 100% TMF has a similar trend of increasing depending on the beating degree (Figure 4a). TMF reaches the maximum tensile index value of 80 N.m·g$^{-1}$ at PFI 6000 revolution, which is about 20% below other samples. TMF indicates a lower tensile index compared to cellulose nanopaper 86–101 N.m·g$^{-1}$ [5]. Awada et al. [1] found a 31% decrease in the breaking length of chemically modified fibers.

The bursting index, a crucial characteristic of paper, increases proportionally with the degree of beating. The papers obtained in the study reached their peak at approximately 6000 PFI revolutions. However, the unexpected impact of beating TEMPO fibers on the bursting index remained consistent with the levels observed in unbeaten pulp (Figure 4b). The tearing index of paper diminishes as the beating process progresses, with a significant

drop of nearly 50% observed after 8000 PFI revolutions beating, as illustrated in Figure 4c. Moreover, the tearing index of the sample containing TMF is notably low, approximately 90% less than that of other samples. Awada et al. [1] reported a 23.9% decrease in the tearing index for chemically modified fibers. While the addition of modified fibers initially shows a slightly positive effect on the tearing resistance of unbeaten pulp (Figure 4c), this effect becomes negative as the beating continues. Bending stiffness expresses the ability of a paper or board strip to resist a bending force applied perpendicularly to the free end of a strip clamped at the other end using the two-point loading method. According to the results (Figure 4d), the maximum values are the same for the NaOH + urea 3% and 1% groups. The values are between 112 and 129 mN. There is one distinctive result in stiffness values in the TMF group. The sheets that have TEMPO modification without beating have a value of 132 mN.

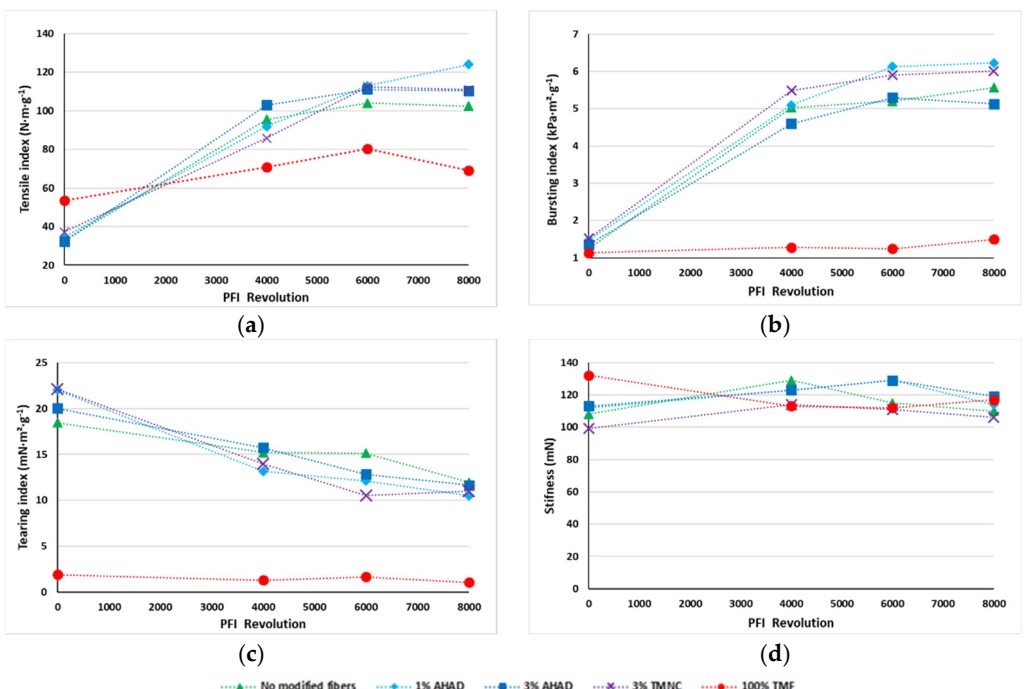

**Figure 4.** The mechanical strength properties of paper sheets: (**a**) tensile index of samples, (**b**) bursting index of samples, (**c**) tearing index of samples, (**d**) stiffness values of samples.

Optical characteristics are important for bleached papers. Figure 5a–d illustrate the negative impact of TEMPO modification on the optical properties of fiber. The decrease in ISO brightness to 67% is a significant loss. Similarly, the CIE whiteness and opacity values of TMF are lower than those of the other papers.

TEMPO oxidation causes degradation and yellowing of fibers, and yellowness values increase from 6.5 to above 20 (Figure 5d).

The addition of modified fibers to paper furnishings causes changes in the color properties of papers (Figure 6a–d). Similarly, for other optical properties, the TMF sheets are negatively differentiated from the other samples, while the color differences ΔE increase from about one to eight points for beaten pulps (Figure 6d). High beating levels increase the color difference by about 2 points, while the color difference in TMF sheets increases by up to 9.3 points. Loosening of the fiber wall or reducing the bending stiffness of the fiber wall due to a decrease in the effective E-modulus occurs as a result of internal fibrillation. Flexibility is a key factor as it governs the most physical and optical properties of pulp and paper, including paper formation and paper strength [19].

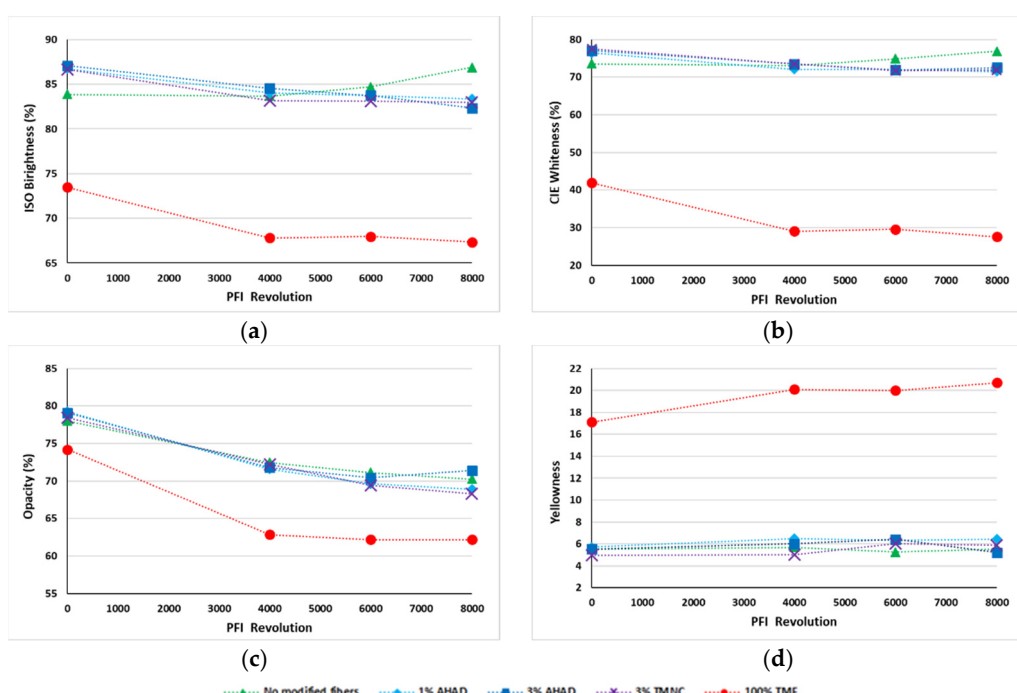

**Figure 5.** Optical properties of paper sheets: (**a**) ISO brightness, (**b**) CIE whiteness, (**c**) opacity, (**d**) yellowness.

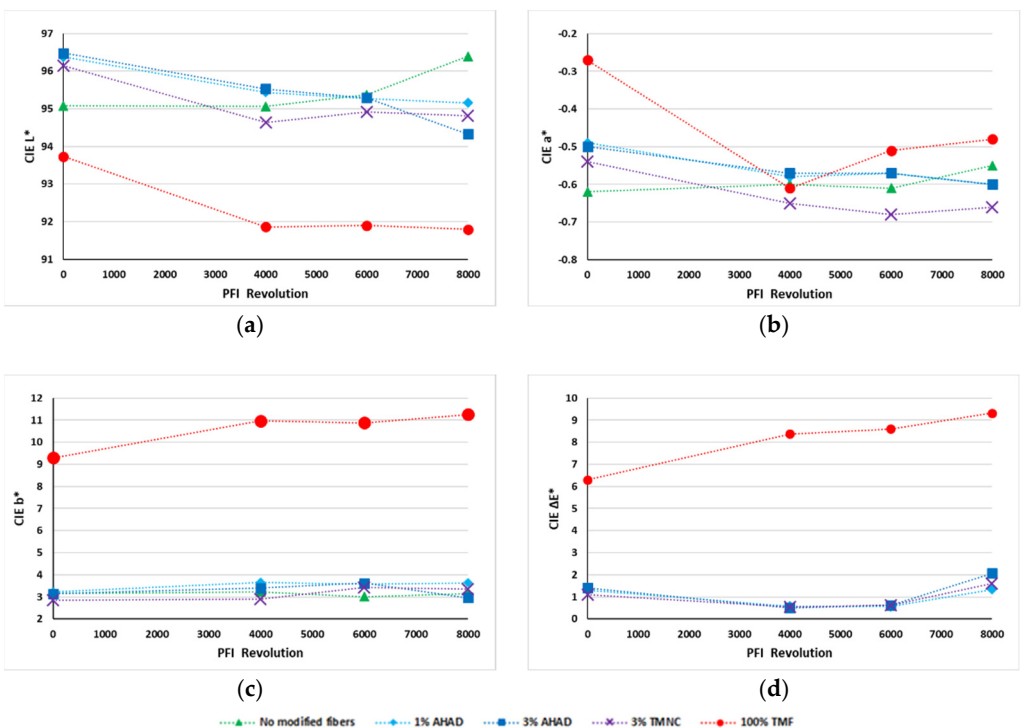

**Figure 6.** CIE L (**a**), CIE a* (**b**), CIE b* (**c**) and ΔE* (**d**) color values of paper sheets.

According to TMA data (Figure 7a,b), the in-plane maximum contraction of samples is concentrated in a region from −0.35% to −0.45% and temperature from 132 to 146 °C. Similarly, for other properties, TMF differs with an average contraction of about −0.59% and a temperature of 168 °C. Paper structure (arrangement of fibers) determines some of the properties of paper in different directions. The cellulose chains are laid with a low angle (relatively parallel) to a fiber axis, and it is expected to be affected in an out-of-plane direction by moisture more than in an in-plane direction. Figure 7b illustrates the maximum

contraction of samples in the out-of-plane direction. All values were below −4%, and no grouping (clustering) was observed for different samples.

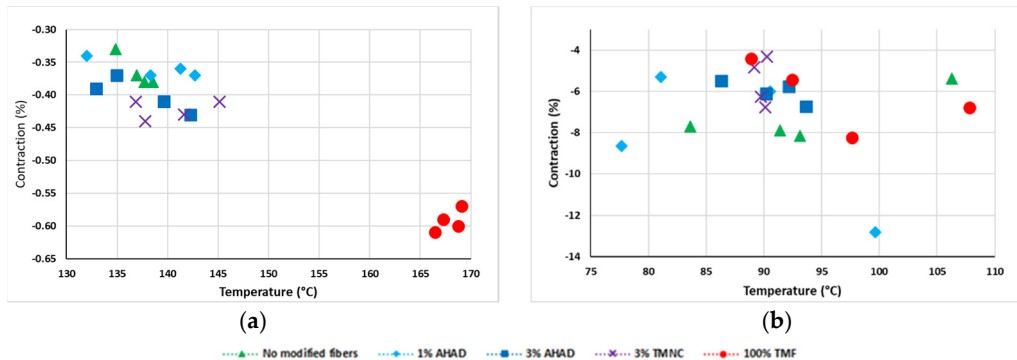

**Figure 7.** The maximum in-plane (**a**) and out-of-plane (**b**) contraction ratios and temperatures.

The coefficient of thermal contraction increases sharply until 40 °C and reaches its maximum value at approximately 45 °C. In the continuation, contraction decreases at a high rate at the beginning and more steadily after 120 °C (Figure 8a). Two samples containing TMF reached the highest contraction rate [43].

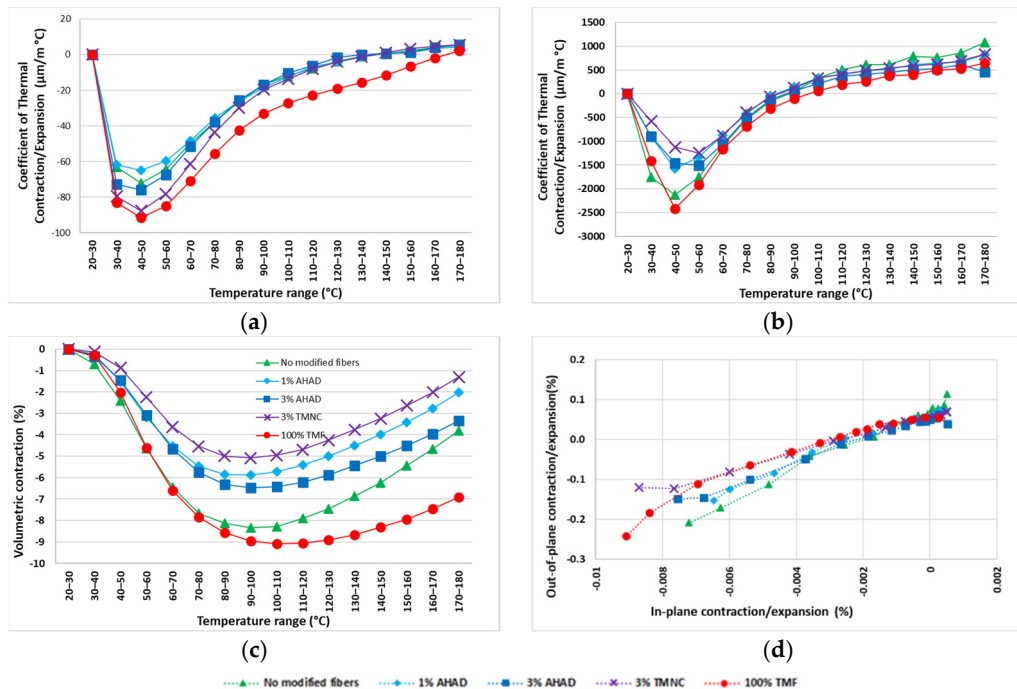

**Figure 8.** In-plane (**a**) and out-of-plane (**b**) coefficient of thermal contraction/expansion of 6000 PFI beaten samples, (**c**) volumetric contraction of 6000 PFI beaten samples, (**d**) comparison of out-of-plane and in-plane contraction rates.

The out-of-plane contraction rate indicates similar contraction trends (Figure 8b). The contraction coefficient has its maximum at approximately 45 °C and decreases thereafter. Near 100 °C, samples reach the initial dimension, and expansion commences. TMF sheets have substantial contraction, while the paper with no modification exhibits the highest expansion coefficient at 180 °C.

Figure 8c shows the volumetric changes during the thermal examination of the samples. TMF demonstrates the highest volumetric contraction of 9%, while the volumetric

contraction of 3% TMNC is only 5%. Paper samples have a maximum contraction between 90 and 110 °C.

The contraction rates of samples in the in-plane and out-of-plane directions differ (Figure 8d). The out-of-plane contraction rate peaks and then returns to the initial value, indicating faster recovery, followed by expansion. Meanwhile, the in-plane contraction is recorded at $-0.002\%$. The expansion of in-plane samples was found to be limited. All paper samples have the same thermal contraction and expansion behavior.

Based on the findings, papers manufactured from TMF exhibit inferior mechanical and optical properties. The main reason is the shortening of the chain by the oxidation of cellulose. Qin et al. [44] reported that during the oxidizing reaction, some of the amorphous region in the cellulose fiber was modified and gradually hydrolyzed, but the crystalline region remained intact. The oxidation medium freely penetrates through the amorphous structure, leading to a higher formation of the carboxyl group in this region [45]. Similar to the literature, since the chemical treatment enhances the amorphous regions, it resulted in TEMPO-modified fibers with a higher crystallinity index, and, concomitantly, a higher thermal contraction coefficient. The tempo-oxidation process weakens the amorphous region of fibers, resulting in an increase in contraction in the axial (in-plane) direction of samples.

## 4. Conclusions

In this study, we explored the impact of various fiber modifications and beating degrees on the properties of paper sheets. Our comprehensive analysis revealed several key insights:

The addition of modified fibers, except 100% TEMPO-modified fibers (TMF), generally led to improved paper density and certain strength properties, particularly at low beating degrees. This suggests the potential of using chemically modified fibers to enhance paper strength.

FTIR analysis was employed to investigate potential alterations, such as chemical changes in the material's structure and crystallinity modifications, and their impact on thermo-mechanical properties. In the corresponding FTIR spectra of TMF sheets, the carboxylate formation of stretching at 1598 c m$^{-1}$ and an increase in the lateral order crystallinity index were observed. A degradation occurred in the amorphous region, as stated in the literature, indicating the oxidation of the amorphous region occurs when the PEMO-mediated oxidation method is used [46,47].

However, the incorporation of TEMPO-modified fibers had a detrimental effect on the optical properties, including brightness and whiteness, which significantly decreased. The presence of carboxylate groups and the degradation of fibers contributed to this outcome.

Thermal analysis revealed that in-plane contraction ratios increased with the addition of TEMPO-modified fibers. All samples showed the maximum coefficient of contraction at about 40–50 °C. Volumetric contraction increased with the addition of modified fibers compared to nonmodified and 100% TMF samples. The amorphous regions of these fibers appeared to undergo more degradation, causing them to contact more than other fiber types.

Our findings suggest that the choice of fiber modification and beating degree should be carefully considered in paper production to balance strength and optical quality. Mild chemical modifications, apart from TEMPO, may offer an avenue for improving paper strength, while the in-plane contraction ratios should serve as a useful indicator of amorphous region destruction.

In conclusion, this study sheds light on the complex interplay between fiber modification, beating degree, and paper properties. It underscores the need for a nuanced approach to paper manufacturing, taking into account both strength and optical characteristics. Future research in this area may further refine our understanding of these relationships and provide insights into optimizing paper production processes.

**Author Contributions:** Conceptualization, A.G.T. and C.A.; methodology, A.G.T. and J.B.; validation, C.A.; writing—original draft preparation, A.G.T. and C.A; supervision, P.O. and I.P.; project administration, P.O. All authors have read and agreed to the published version of the manuscript.

**Funding:** This research was funded by Istanbul Üniversitesi-CERRAHPASA, grant number 31746, and the APC was funded by ARIS Slovenian Research and Innovation Agency within the program P4-0015.

**Data Availability Statement:** Data sharing not applicable.

**Acknowledgments:** The authors acknowledge the Papirnica Vevče d.o.o., University of Ljubljana, and the Council for Scientific Research of Istanbul University-Cerrahpaşa for supporting this work.

**Conflicts of Interest:** The authors declare no conflict of interest.

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
