# Peer review of "Effects of Modified Cellulose Fiber and Nanofibril Integration on Basic and Thermo-Mechanical Properties of Paper"

_forests, doi:10.3390/f14112150_

Round 1

Reviewer 1 Report

Comments and Suggestions for Authors

The experimental article “The thermo-mechanical, physical, and optical properties of paper obtained by addition of modified cellulose fibers and nanofibrils” is devoted to the results of a study of the effect of adding modified fibers on the properties of paper and in all formal respects corresponds to the Forests publication. The topic of paper modification is very relevant at the present time, so the presented results deserve the attention of specialists in the field of cellulose and its modification products in order to search for new areas of application. The authors made a selection of cellulose types (coniferous fibers, specifically spruce, pine and larch) and cellulose modifications, and comprehensively studied the change in the properties of paper sheets. The article is illustrated with 21 drawings, the quality of which requires additional work. Scientific novelty, at first glance, is not obvious, but the scientific hypothesis is visible in the statement that any cellulose can be a source for modification products that significantly change the properties of composites, and the degree of grinding of these fibers determines the performance characteristics of paper. The use of the term “beating” interferes with the perception of the results obtained, since it is unclear how this procedure affected, for example, the length of the fibers. It would be much better if generally accepted terms were used. The article requires certain changes, a list of which is given below.

Notes:

1. Please note that the list of paper properties in the title, annotations and further in the text is different. Bring into compliance with minimal changes.

2. Edit the introduction, since the presentation does not have a clear sequence. For example, after describing the functionalization of nanocellulose, the text begins about the preparation of nanocellulose and its use as an independent component (lines 101-110).

3. Please note that the introduction cites publications 1-21, among which the 2023 edition is not included.

4. In section “2.1. Materials and sheet making” additionally and clearly indicate to which cellulose the “additive” was added. It's not written very clearly.

5. In section “2.2. Sheet characterization" in addition to the ISO listings, provide links to recent publications of authors who used the same methods in their research.

6. Please note that the discussion of each figure must be accompanied by a rationale for the observed phenomenon. The way it is presented in this version is clearly not enough. It seems that the result is obvious and predictable.

7. Figure 4. Stiffness properties of papers. Provide more explanation for the observed chaos of the results. Compare with previously published data.

8. When describing Figures 6 and 7, there is no main conclusion that “beating” has little effect on optical performance. The bottleneck in the manuscript.

9. It is recommended to combine some of the figures into one with a single legend, since the given graphs are not very informative, and the description of them does not reveal the scientific novelty of the article.

10. In this case, the figure itself “Figure 14. FTIR absorbance spectrum of papers” and the description for it can be placed in Saplimentary, since this information does not develop the topic indicated by the title.

11. More discussion of the Figure 15. Lateral Order Crystallinity Index A1430/A897 data is necessary, since it is highly doubtful that “beating” contributes to an increase in this indicator. Give examples of such studies and compare your own data with published ones.

12. It is necessary to compare the results in Figures 16 and 17 with other authors and provide links.

13. Figures 18-20 should be combined into one figure and the observed phenomena should be carefully described with references to published works.

14. Lines 314-315. The offer should be checked. The statement that modified cellulose has greater crystallinity than the original cellulose is erroneous.

15. Section 4. Conclusions edit, in particular, “FTIR spectra of TMF sheets showed” cannot indicate something... The degree of oxidation, determined by a known method, may indicate a violation of the structure of the original cellulose. By the way, this data is not in the manuscript. Add data.

Reviewer 2 Report

Comments and Suggestions for Authors

The topic of this work is relevant to the field of the Forests journal. Manuscript originality is not high, which does not diminish the substantive value of the work. The work certainly brings new knowledge to the subject area. Does the topic address a specific gap in the field? - The authors did not indicate this clearly, it should be supplemented in the text. However, the discussion should be more refined and based on a comparison of the work with the research of other authors on this topic. It should be indicated the innovative aspect of the work and the elements that distinguish it from the achievements of other researchers in this field.

Moreover, further improvements are recommended, as follows:

1. The abstract needs to be modified, it is too extensive. A good abstract should include: a) Background (highlight the purpose of the study); b) Methods (describe briefly the main methods or treatments applied); c) Results (summarize the article's main findings); and d) Conclusion (indicate the main conclusions). Detailed numerical data from research results should not be included here.

2. In the MATERIALS AND METHODS section, it would be good to provide the equipment on which individual determinations were made along with the necessary information (manufacturer, city, country).

3. Figures - axes descriptions and legend are poorly visible, the font should be modified so that there are no problems with the legibility of the images.

Other than this, the manuscript is well-prepared, which will provide meaningful insights on future related work. I would like to recommend its acceptance for publication after the authors address the above-mentioned issues.
